# Therapeutic Monoclonal Antibodies Targeting Immune Checkpoints for the Treatment of Solid Tumors

**DOI:** 10.3390/antib8040051

**Published:** 2019-10-21

**Authors:** Nicholas Gravbrot, Kacy Gilbert-Gard, Paras Mehta, Yarah Ghotmi, Madhulika Banerjee, Christopher Mazis, Srinath Sundararajan

**Affiliations:** 1Division of Hematology-Oncology, Department of Medicine, University of Arizona Cancer Center, Tucson, AZ 85724, USA; parasmehta@email.arizona.edu (P.M.); yarahghotmi@email.arizona.edu (Y.G.); mbanerje@email.arizona.edu (M.B.); mazisc@email.arizona.edu (C.M.); srinathshri@gmail.com (S.S.); 2Texas Oncology, Dallas, TX 75251, USA

**Keywords:** immunotherapy, checkpoint inhibitor, monoclonal antibody, cancer therapy

## Abstract

Recently, modulation of immune checkpoints has risen to prominence as a means to treat a number of solid malignancies, given the durable response seen in many patients and improved side effect profile compared to conventional chemotherapeutic agents. Several classes of immune checkpoint modulators have been developed. Here, we review current monoclonal antibodies directed against immune checkpoints that are employed in practice today. We discuss the history, mechanism, indications, and clinical data for each class of therapies. Furthermore, we review the challenges to durable tumor responses that are seen in some patients and discuss possible interventions to circumvent these barriers.

## 1. Introduction

In recent years, the limitations of conventional chemotherapy have spurred research into more precise cancer treatment, using targeted therapies in hopes of selectively eradicating cancer while sparing normal host cells. As new cancer cell markers, cytokines, and immunologic checkpoints have been discovered, monoclonal antibodies (mAb) and small molecule inhibitors have been developed to accomplish these goals. An important discovery in this area is that of immune checkpoint molecules, which dampen anti-cancer immune responses. Such proteins include programmed cell death protein-1 (PD-1), its ligands programmed death-ligand 1 and 2 (PD-L1, PD-L2) and cytotoxic T-lymphocyte-associated protein 4 (CTLA-4), among others. Inhibitors of immune checkpoints have since been developed as a means to “take the breaks off” of an otherwise impeded anti-cancer immune response. As additional targets have been discovered, new therapies have emerged. Herein, we review current mAbs directed at immune checkpoint modulation within the context of treating various solid cancers. 

## 2. Immunotherapy Overview

### 2.1. Overview

Several classes of checkpoint modulators have been studied clinically. A summary of those with US Food and Drug Administration (FDA) approval is provided in Table A1. Table A1 additionally includes immunologic and pharmacologic parameters, such as IgG subtype, target affinity, and epitope properties, for each drug. Figure 1 provides an overview of implicated receptor-ligand interactions and their generalizable effects on the immune response.

For some of the mAbs discussed (such as CTLA-4 inhibitors), the exact mechanism of action is not fully understood and is the subject of active investigation. Therefore, the mechanisms presented represent the basic foundation of a presumably more complicated biochemical picture.

### 2.2. CTLA-4 Inhibitors 

#### 2.2.1. Background 

Discovered in 1987, CTLA-4 was identified as a homolog of CD28, and later, an inhibitor to T cell activation [1,2]. After several years of preclinical and clinical trials, the first CTLA-4 inhibiting mAb, ipilimumab (Yervoy^®^; BMS-734016; MDX-010; Bristol-Myers Squibb), gained FDA approval in 2011 for the treatment of unresectable or metastatic melanoma. A second CTLA-4 mAb, tremelimumab (CP-675,206; AstraZeneca), has also been developed.

#### 2.2.2. Mechanism of Action 

CTLA-4 is a homolog of CD28 with opposing functions. Both are expressed by T cells and bind the ligands B7-1 and B7-2 found on antigen-presenting cells (APC). When CD28 binds B7-1 and B7-2, intracellular signaling via phosphatidyl-inositol 3-kinase (PI3K) results in downstream activation of transcriptional factors that increase proliferation, differentiation, and survival of T cells [3]. The binding of B7-1 or B7-2 to CTLA-4, however, prohibits this response. CTLA-4 has a higher affinity for B7-1/B7-2 ligands relative to CD28, and as such, it can outcompete CD28, allowing for suboptimal stimulation of T cells. CTLA-4 is constitutively expressed on regulatory T cells (Treg) and plays an important role in immune system suppression.

Ipilimumab and tremelimumab are monoclonal IgG antibodies targeted against CTLA-4. Ipilimumab is of the IgG1 subclass, whereas tremelimumab is an IgG2 molecule. Both have similar binding affinity to CTLA-4, though ipilimumab has a higher dissociation rate. Epitopes are also comparable between mAbs, as both bind to the F and G strands of the CTLA-4 molecule [4]. The inhibitory effect of either mAb against CTLA-4 facilitates increased CD28/B7 binding, thus upregulating T cell proliferation and immune activity [5,6]. A proposed secondary mechanism of CTLA-4 mAbs is that of Treg depletion within the tumor microenvironment (TME), as some studies have shown that decrease in Tregs with anti-CTLA-4 therapy [7,8,9,10]. However, several other studies have reported data that contradicts these findings, with no evidence of Treg depletion [11,12,13,14]. Therefore, this mechanism is still under investigation.

#### 2.2.3. Indications

Currently, ipilimumab is the only anti-CTLA-4 mAb with FDA approval. It was first approved in 2011 for the treatment of unresectable or metastatic melanoma, and since then, its indications have expanded significantly (Table A1). Tremelimumab is undergoing investigation in various solid malignancies but has not yet been granted FDA approval so far. Figure 2 provides a timeline of noteworthy changes in FDA approvals for ipilimumab. 

### 2.3. PD-1 Inhibitors 

#### 2.3.1. Background 

The first phase I trial investigating anti-PD-1 mAbs was reported in 2012, and the field has grown immensely since then [15]. Nivolumab (Opdivo^®^; BMS-936558; Bristol-Myers Squibb) and pembrolizumab (Keytruda^®^; MK-3475; Merck) represent the most well-studied PD-1 inhibitors, though several other mAbs within this class exist. 

#### 2.3.2. Mechanism of Action

PD-1 is a type I membrane protein that is loosely related to CTLA-4 structurally [16]. It is expressed on activated T cells, B cells, and macrophages [17]. Under physiologic conditions, PD-1 negatively regulates T cell activity to maintain peripheral immune tolerance and to avoid immunopathology. It accomplishes this by binding to its ligands, PD-L1 and PD-L2, resulting in activation of an intracellular phosphatase, which, in turn, inhibits downstream kinase signaling customarily involved in T cell activation. Therefore, this results in decreased T cell proliferation and response [18,19].

As with CTLA-4 inhibitors, PD-1 inhibitors work by enhancing the patient’s natural anti-tumor immune response. IgG mAbs that block PD-1, such as nivolumab and pembrolizumab, inhibit the intracellular signaling cascade described above, resulting in disinhibition of the anti-tumor immune response. PD-1 inhibitors also block interactions with PD-L2. It should be noted that this is an advantage over PD-L1 inhibitors, which do not block PD-1/PD-L2 interactions [20]. 

Both nivolumab and pembrolizumab are of the IgG4 subclass, but the epitope binding regions of each differ. For nivolumab, its epitope is predominantly in the PD-1 N-loop, whereas pembrolizumab’s epitope primarily involves the PD-1 CD loop. Both have high affinity and high specificity for PD-1 [21].

#### 2.3.3. Indications 

In late 2014, both nivolumab and pembrolizumab received their first FDA-approved indications for use in unresectable or metastatic melanoma in patients with history of disease progression following ipilimumab and, if BRAF V600 mutation positive, BRAF inhibition. Since then, the scope of anti-PD-1 therapy has broadened tremendously (Table A1). A timeline of changes in FDA-approved indications is provided in Figure 3 for nivolumab and Figure 4 for pembrolizumab.

### 2.4. PD-L1 Inhibitors 

#### 2.4.1. Background 

While anti-PD-1 therapies “blazed the trail” for modulation of the PD-1/PD-L1 axis, research into mAbs directed against PD-L1 followed shortly after. Atezolizumab (Tecentriq^®^; Genetech/Roche) has garnered the most investigation within the class, but others include durvalumab (Imfinzi^®^; Medimmune/AstraZeneca) and avelumab (Bavencio^®^; Merck/Pfizer). 

#### 2.4.2. Mechanism of Action 

PD-L1 is a type I transmembrane protein expressed by T cells, B cells, natural killer (NK) cells, macrophages, dendritic cells, and epithelial cells; pathologically, PD-L1 can directly be expressed by cancer cells [22]. The effects of PD-L1/PD-1 interactions have been discussed previously. Like PD-1 inhibitors, PD-L1 inhibitors prevent ligand-receptor binding, blocking the immune suppressive effects mediated through this pathway. In addition to PD-1/PD-L1 interactions, PD-L1 is also known to bind competitively to B7-1, with similar effects as CTLA-4/B7-1 binding. Thus, PD-L1 inhibitors also block interactions between PD-L1 and B7-1, which further disinhibits anti-cancer immunity [20].

Atezolizumab, avelumab, and durvalumab are all IgG1 mAbs [23]. Each has a different epitope but share key interactions within PD-L1’s CC’FG β-sheet. Unique interactions involve regions within the BC, CC’, C’C’’ and FG loops (atezolizumab); CC’ loop and N-terminus (durvalumab); and CC’ loop (avelumab). All have been shown to have dissociation constants of less than 1 nM, indicating high affinity for PD-L1.

#### 2.4.3. Indications

The first PD-L1 mAb to be granted FDA approval was atezolizumab, which received accelerated approval in May 2016 for the treatment of locally advanced or metastatic urothelial cancer resistant to platinum-containing chemotherapy regimens. Several other approvals have been granted over time (Table A1). Figure 5 provides a summary of changes in FDA-approved indications for the various PD-L1 mAbs.

### 2.5. LAG-3 Inhibitors

#### 2.5.1. Background

While lymphocyte-activation gene-3 (LAG-3) modulation dates back as far as 2006 with the LAG-3-immunoglobulin fusion protein eftilagimod alpha (IMP321; Immutep), the earliest mAb directed against LAG-3 was relatlimab (BMS-986016; Bristol-Myers Squibb), which was first evaluated in the treatment of several solid malignancies from 2013–2017 [24]. Relatlimab continues to be the subject of interest in several active clinical trials. 

#### 2.5.2. Mechanism of Action 

LAG-3 is a type I transmembrane protein expressed by activated T cells, B cells, NK cells and dendritic cells, and it is involved in negative regulation of helper and cytotoxic T cell responses [25,26,27]. Activation of LAG-3 occurs peripherally via binding to class II major histocompatibility complex (MHC) and/or galectin-3 [27]. Activated LAG-3 reduces production of various immunostimulatory interleukins (IL) and enhances sensitivity to Treg signaling, thus increasing tolerance and accelerating T cell exhaustion [27,28]. Anti-LAG-3 antibodies prevent binding of LAG-3 with its ligands, blocking these effects and thereby facilitating increased anti-tumor activity.

#### 2.5.3. Indications 

There are no FDA-approved indications thus far for anti-LAG-3 mAbs. Investigations in various solid and hematologic malignancies are ongoing, with a phase II/III study representing the highest stage of development. 

### 2.6. TIM-3 Inhibitors

#### 2.6.1. Background

T cell immunoglobulin and mucin-domain containing-3 (TIM-3) mAbs are currently under investigation in early clinical studies, with the first phase I trial having opened in 2015. Agents in this class include Sym023 (Symphogen), TSR-022 (Tesaro/AnaptysBio), MBG453 (Novartis), LY3321367 (Lilly), and BGB-A425 (BeiGene).

#### 2.6.2. Mechanism of Action 

TIM-3 is a type I transmembrane protein implicated in suppression of T cell-mediated immune responses. TIM-3 downregulates the production of several cytokines, including IL-2, IL-12, interferon-beta, and interferon-gamma, and additionally expedites T cell exhaustion [29,30]. Given the possibility for expression in multiple cell lines, TIM-3 may dampen T cell activation at varying stages of the immune response, even upstream to direct T cell interactions [29]. TIM-3 inhibitors block these responses, mitigating the immune inhibition mediated through this pathway. As with the other checkpoint inhibitors, this disinhibition results in an enhanced anti-tumor immune response. 

#### 2.6.3. Indications

There are no current FDA-approved indications for anti-TIM-3 mAbs. A number of early studies assessing anti-TIM-3 therapy in localized or advanced solid malignancies are ongoing.

### 2.7. CD40 Agonists

#### 2.7.1. Background

Immune checkpoint blockade is often ineffective when severe immunosuppression develops; thus, targets that further promote an intratumoral T cell response are needed [31]. One such target is CD40, a member of the tumor necrosis factor (TNF) receptor superfamily expressed on both immune and non-immune cells [32]. Therapies within this class include CP-870,893 (RG-7876; Pfizer/Roche) and APX005M (Apexigen).

#### 2.7.2. Mechanism of Action 

CD40 is a cell surface molecule of the TNF receptor superfamily that is present physiologically on APCs and pathologically on tumor cells [33]. The CD40 ligand (CD40L/CD154) is expressed on activated CD4+ and CD8+ T cells, B cells and NK cells [34]. CD40/CD40L signaling serves as the bridge between innate and adaptive immunity; CD40 expression on APCs enhances antigen presentation and co-stimulatory capacity, resulting in a robust activation of cytotoxic T cells even in the absence of a CD4+ T cell helper signal [34]. Thus, mAbs acting as agonists for CD40 result in vigorous immunologic activation and proliferation.

#### 2.7.3. Indications 

There are no current FDA-approved indications for CD40 agonists. A number of early studies are ongoing. 

### 2.8. OX40 Agonists 

#### 2.8.1. Background 

Another member of the TNF receptor superfamily is OX40 (CD134), which has similar functions. Therapeutic OX40 agonists include 9B12 (Providence Health, Renton, WA, USA). 

#### 2.8.2. Mechanism of Action

OX40 is a type I transmembrane glycoprotein expressed on activated T cells; it is a member of the TNF receptor superfamily. In much the same fashion as CD40L, OX40 ligand (OX40L) is expressed at low levels in cells throughout the body under physiologic conditions and is upregulated in inflammatory conditions such as autoimmune processes [35]. The OX40/OX40L interaction involves several downstream signaling cascades that promote cell survival and enhance cytokine release from T cells [36,37]. By leveraging these pathways, OX40 agonists have the potential to stimulate the immune response for use against cancer. 

#### 2.8.3. Indications 

There are no current FDA-approved indications for anti-OX40 antibodies. Several phase I/II studies assessing anti-OX40 therapy as monotherapy or combined therapy for solid malignancies are ongoing.

## 3. Clinical Evidence 

### 3.1. Overview

Given the vast number of studies published in checkpoint modulation, a detailed account of each trial is beyond the scope of this review. Herein, we discuss the most clinically relevant studies, including those that led to FDA approval and/or changes in standards of care. More information about studies not covered in the text can be found in each class’s respective table(s), alongside summaries of the studies discussed here.

### 3.2. CTLA-4 Inhibitors

#### 3.2.1. Ipilimumab 

##### Overview 

Table A2 summarizes current clinical data for CTLA-4 inhibitors.

##### Melanoma 

Ipilimumab was granted its first FDA approval in 2011 for unresectable/metastatic melanoma. This came about following completion of a phase III trial evaluating ipilimumab alone versus in combination with a gp100 cancer vaccine in patients with unresectable, refractory stage III or IV melanoma [38]. Numerous studies followed, exploring the role of ipilimumab both alone and in combination with other treatment modalities in varying stages of melanoma. Details of these investigations can be found in Table A2. Ipilimumab has since had its FDA approval expanded to include adjuvant treatment following complete surgical resection in patients with stage III disease.

##### Renal Cell Carcinoma

The phase III CheckMate 214 trial explored the efficacy of ipilimumab plus nivolumab for renal cell carcinoma (RCC), comparing it with first-line standard of care sunitinib. Results from this trial demonstrated significantly higher 18-month overall survival (OS) with combination ipilimumab/nivolumab compared with sunitinib (hazard ratio (HR) = 0.63, *p* < 0.001), as well as higher objective response rate (ORR) (*p* < 0.001) in the combination group [39]. Consequently, in April 2018, the FDA approved the use of combination ipilimumab/nivolumab for previously untreated, intermediate- to poor-risk advanced RCC. Long-term follow up data (median follow up 32.4 months) published recently showed that in intermediate-risk or poor-risk patients, combination ipilimumab/nivolumab continued to be superior to sunitinib in terms of OS [40].

##### Non-Small Cell Lung Cancer 

Encouraged by early clinical trials showing ipilimumab’s activity against a variety of tumor types, researchers began investigating its use in patients with non-small cell lung cancer (NSCLC). However, ipilimumab alone or as part of combination therapy has not yielded meaningful clinical benefit and is not approved for lung cancer treatment [41,42].

#### 3.2.2. Tremelimumab

While tremelimumab received orphan drug status for treatment of mesothelioma in 2015, it has yet to be FDA-approved for this indication. The DETERMINE study found no significant life prolongation in patients with previously treated malignant mesothelioma who were given tremelimumab, compared to those given placebo, with a median OS of 7.7 months in the treatment group and 7.3 months in the placebo group (HR = 0.92, *p* = 0.41) [43].

### 3.3. PD-1 Inhibitors 

#### 3.3.1. Nivolumab

##### Overview 

Table A3 summarizes current clinical data for nivolumab.

##### Bladder Cancer

In 2017, nivolumab received FDA approval as second-line monotherapy for metastatic or surgically unresectable urothelial carcinoma that had progressed or recurred despite prior treatment with at least one platinum-based chemotherapy regimen. This was granted in response to a phase II clinical trial in which Sharma et al. treated 270 patients with metastatic urothelial carcinoma using a regimen of nivolumab 3 mg/kg IV every two weeks. ORR was 19.6% across all PD-L1 expression subgroups, with 2% experiencing complete response (CR), and median progression-free survival (PFS) of 2.0 months. Grade 3 or 4 AEs occurred in 18% of patients treated, consisting mostly of grade 3 diarrhea and fatigue [44].

##### Colorectal Cancer 

In 2017, Overman et al. published results from a phase II trial exploring the use of nivolumab monotherapy in patients with metastatic DNA mismatch repair-deficient (dMMR)/microsatellite instability-high (MSI-H) colorectal cancer (CRC). Patients included in the study had disease progression on, or after, at least one previous line of treatment, including a fluoropyrimidine and oxaliplatin or irinotecan. ORR was 31.1%, with a median PFS of 14.3 months and 12-month PFS rate of 50%. Grade 3 or 4 AEs were noted in 20% of patients [45]. These findings resulted in accelerated FDA approval in July 2017 for treatment of metastatic dMMR/MSI-H CRC that had progressed after treatment with the above chemotherapies.

##### Head and Neck Squamous Cell Carcinoma

In a randomized, open-label, phase III trial, Ferris et al. explored the use of nivolumab monotherapy in patients with recurrent or metastatic head and neck squamous cell carcinoma (HNSCC) that had progressed within six months after platinum-based chemotherapy. Patients were treated with either nivolumab or the investigator’s choice of standard, single-agent systemic therapy (methotrexate, docetaxel or cetuximab). ORR was 13.3% in the nivolumab-treated patients (2.50% complete response rate (CRR)), versus 5.8% in the patients treated with standard therapy (0.83% CRR). Median OS was 7.5 months and 5.1 months in the nivolumab-treated versus standard treatment groups, respectively (HR = 0.70, *p* = 0.01). Fewer grade 3 or 4 adverse events (AE) were reported in the nivolumab-treated group than in the standard chemotherapy group [46]. As a result of this study, nivolumab received FDA approval in November 2016 for recurrent HNSCC that had progressed on or after the above chemotherapies.

##### Hepatocellular Carcinoma 

In a phase II open-label study, El-Khoueiry et al. explored the safety and efficacy of nivolumab monotherapy in the treatment of advanced hepatocellular carcinoma (HCC) with or without chronic viral hepatitis. Overall, ORR was 20% and median PFS was 4.0 months. The ORR was comparable regardless of previous sorafenib treatment. Grade 3 or 4 AEs were noted in 19% of patients with no treatment-related deaths. Of note, PD-L1 expression did not appear to have a significant effect on response rates [47]. Based on these findings, nivolumab was granted FDA approval for treatment of HCC in patients who had failed vascular endothelial growth factor (VEGF) inhibition previously.

##### Hodgkin Lymphoma 

Reed-Sternberg cells are known to utilize PD-L1 and PD-L2 to evade immune surveillance [48]. In response to promising results from a 2014 phase I study, a follow up phase II study was conducted that assessed the clinical benefit and safety of nivolumab in patients with Hodgkin lymphoma (HL) who had failed both autologous stem-cell transplantation (ASCT) and brentuximab vedotin [49]. ORR was 66.3% (9% CRR) and median PFS was 10.0 months; grade 3 or 4 drug-related AEs occurred in 25% of patients, most commonly consisting of increased lipase and neutropenia [50]. These two studies were the bases of accelerated FDA approval of nivolumab in treatment of classical HL (cHL) with progression following ASCT and post-transplant brentuximab.

##### Melanoma 

Based on the results of the phase III CheckMate 037 trial, nivolumab received its first FDA-approved indication in December 2014 for the treatment of unresectable/metastatic melanoma after failure of ipilimumab and, if BRAF V600 mutation positive, a BRAF inhibitor [51]. Like ipilimumab before it, several studies followed to investigate an expanded role for nivolumab in the management of melanoma. Current indications include combination therapy with ipilimumab for BRAF V600 wild-type, unresectable/metastatic melanoma and adjuvant therapy following complete surgical resection for patients with stage III melanoma. Details for each study can be found in Table A3.

##### Non-Small Cell Lung Cancer 

The CheckMate 017 and CheckMate 057 phase III trials assessed nivolumab’s role in treatment of advanced squamous NSCLC (sqNSCLC) and non-squamous NSCLC (NsqNSCLC), respectively [52,53]. In each, nivolumab monotherapy was compared to docetaxel following disease progression after one line of platinum-based chemotherapy. Median OS and ORR were higher in the nivolumab group in both trials, along with longer PFS for patients with advanced sqNSCLC. These studies were the bases of two FDA approvals in 2015 for the use of nivolumab in treatment of metastatic sqNSCLC and NsqNSCLC that had progressed following after platinum-based chemotherapy. Three year follow up data published in 2018 showed a continued, significant OS benefit compared to docetaxel in advanced NSCLC with (HR = 0.68) or without (HR = 0.70) liver metastases [54].

##### Renal Cell Carcinoma 

In 2015, Motzer et al. reported the results of a randomized, open-label, phase III study comparing the effectiveness and safety of nivolumab monotherapy compared to everolimus in patients with RCC who had received previous treatment with one or two regimens of anti-angiogenic therapy. Between the nivolumab-treated and everolimus-treated groups, the median OS was 25.0 months versus 19.6 months, respectively (HR = 0.73, *p* = 0.0018). The ORR was 25% in the nivolumab group and 5% in the everolimus group (*p* < 0.001). Grade 3 or 4 AEs occurred in 19% of the patients treated with nivolumab and in 37% of patients treated with everolimus [55].

This study was followed by the CheckMate 214 trial, which compared combination ipilimumab/nivolumab to sunitinib for advanced RCC. These findings were discussed in the CTLA-4 section.

##### Small Cell Lung Cancer 

In 2018, the FDA granted nivolumab accelerated approval for third-line treatment of metastatic small cell lung cancer (SCLC) based on the results of the CheckMate 032 trial, which compared nivolumab monotherapy to combination nivolumab and ipilimumab. For nivolumab monotherapy, an ORR of 10% was observed; median OS was 4.4 months. Grade 3 or 4 AEs occurred in 13% of patients. The three combination groups, each with varying doses, had ORR ranges from 19–33%, with grade 3 or 4 toxicity rates of 19–30% [56].

#### 3.3.2. Pembrolizumab

##### Overview 

Table A4 summarizes current clinical data for pembrolizumab.

##### Cervical Cancer 

Pembrolizumab received approval as a second-line treatment for cervical squamous cell carcinoma (CSCC) on the basis of KEYNOTE-158, a Phase Ib trial that enrolled 98 patients who had exhausted first-line therapy. Overall ORR was 12.2%; all responses occurred among patients with PD-L1 positive tumors. In this subset of patients, ORR was 14.6%. Median OS was 9.4 months in the total population and 11.0 months in those with PD-L1 positive tumors [57].

##### Gastric Cancer

On the basis of KEYNOTE-059, pembrolizumab received approval for recurrent, advanced gastric or gastroesophageal junction (GEJ) adenocarcinoma with progression on multiple prior therapies and known tumor PD-L1 expression. This phase II trial investigated the use of pembrolizumab in patients with gastric/GEJ adenocarcinoma previously treated with two or more systemic therapies including fluoropyrimidine and a platinum-based therapy, and a HER2/neu therapy if applicable. ORR was 15.5% in the PD-L1 positive group, and 6.4% in the PD-L1 negative group. The median duration of response was 16.3 months in the PD-L1 positive group and 6.9 months in the PD-L1 negative group [58].

##### Head and Neck Squamous Cell Carcinoma 

The KEYNOTE-012, KEYNOTE-040, and KEYNOTE-048 trials extensively evaluated pembrolizumab in the treatment of HNSCC, and several favorable outcomes were reported [59,60,61]. Current FDA-approved indications include first-line treatment for metastatic/unresectable recurrent HNSCC, both alone for tumors with known PD-L1 expression and in combination with platinum-based chemotherapy and fluorouracil for all patients. More information can be found in Table A4. 

##### Hepatocellular Carcinoma 

The KEYNOTE-224 phase II trial evaluated safety and efficacy of pembrolizumab in patients with HCC who had progressed or been intolerant to sorafenib. Results were published in June 2018 and showed an ORR of 17% with 44% of patients having stable disease, and a six-month OS rate of 77.9%. Safety profile was found to be similar to that seen in previous studies of pembrolizumab [62]. This resulted in a new FDA-approved indication for treatment of HCC after failure of sorafenib.

##### Hodgkin Lymphoma 

The known overexpression of PD-L1 and PD-L2 in HL was the basis of KEYNOTE-087, a phase II trial that evaluated pembrolizumab in three different cohorts stratified by prior treatment history [63]. Overall ORR was 69%, with relatively equal distribution among cohorts. CRR was 22.4%, and 31 patients had durable responses lasting six months or greater. This led to FDA approval of pembrolizumab for refractory or relapsed cHL after three or more treatments.

##### Lung Cancer 

Pembrolizumab has been extensively studied in NSCLC and, more recently, SCLC in refractory and front-line settings. NSCLC represents one of the first FDA-approved indications granted to pembrolizumab, as it received accelerated approval in October 2015 for patients with metastatic NSCLC with known tumoral PD-L1 expression that had progressed on or after platinum-based chemotherapy [64]. It currently has indications as part of combination first-line treatment for metastatic NsqNSCLC and sqNSCLC, as well as first-line treatment for stage III disease in patients who are not candidates for surgical resection or definitive chemoradiation, whose tumors have no epidermal growth factor receptor (EGFR) or anaplastic lymphoma kinase (ALK) mutations, and whose tumors are known to express PD-L1. For SCLC, FDA approval was granted in June 2019 for patients with metastatic SCLC that had progressed on or after platinum-based chemotherapy and at least one other line of treatment. Please see Table A4 for further details of each study.

##### Melanoma 

KEYNOTE-001, KEYNOTE-002, KEYNOTE-006, and KEYNOTE-054 assessed the role of pembrolizumab in management of melanoma in various settings [65,66,67,68]. On the basis of early results from KEYNOTE-001, pembrolizumab was granted its first ever FDA approval in September 2014. Indications have expanded since; they include metastatic melanoma with disease progression on ipilimumab and, if BRAF V600 mutation positive, a BRAF inhibitor, as well as adjuvant treatment following resection for stage III disease. Table A4 summarizes these studies.

##### Merkel Cell Carcinoma 

The efficacy and safety of pembrolizumab in patients with systemic chemotherapy naïve advanced Merkel cell carcinoma (MCC) was assessed in the phase II KEYNOTE-017/Cancer Immunotherapy Trials Network-09 trial, which published final results earlier this year. Subjects received pembrolizumab monotherapy for up to two years, with an ORR of 56%, CRR of 24% and partial response rate (PRR) of 32%. Median PFS was 16.8 months, and two-year OS rate was 68.7%. An association was noted between PD-L1 positive tumors and improved PFS and OS [69]. This study led to the approval of pembrolizumab as a first-line treatment in adult and pediatric locally advanced or metastatic disease.

##### MSI-H or dMMR Tumors (Tissue-Agnostic)

In 2017, the FDA approved pembrolizumab as the first tissue-agnostic cancer therapy for unresectable or metastatic solid cancers expressing MSI-H or dMMR, marking the first FDA approval based on biomarker expression rather than on specific disease. The approval was based off of the data from two main studies, KEYNOTE-016 and KEYNOTE-164, and post hoc analyses of three studies from which MSI-H or dMMR patients were identified: KEYNOTE-012, KEYNOTE-028, and KEYNOTE-158. The data consisted of 135 prospective patients and 14 retrospective patients whose MSI-H and dMMR status was identified using either polymerase chain reaction or immunohistochemistry. There were 90 CRC patients and 59 patients with one of 14 other solid tumor types who were treated with either pembrolizumab 200 mg every three weeks or pembrolizumab 10 mg/kg every two weeks. The results showed an ORR of 39.6%, with 78% of those patients showing a response duration greater than 6 months [70]. The approval currently exists for solid tumors that have progressed after treatment with no other current treatment options, or CRC that has progressed after being treated with fluoropyrimidine, oxaliplatin, and irinotecan. 

##### Primary Mediastinal B-Cell Lymphoma 

The phase I KEYNOTE-013 study [71] and follow-up phase II KEYNOTE-170 trial [71] evaluated pembrolizumab in patients with relapsed/refractory primary mediastinal B-cell lymphoma (rrPMBCL) who had failed, were ineligible for, or refused ASCT. Safety and efficacy data were promising in the phase I component, with similar response in the phase II follow up for patients with rrPMBCL that had relapsed after two or more lines of therapy. Phase II data demonstrated an ORR of 45% and CRR of 13%; at the data cutoff, none of patients showing CR had relapsed. Median PFS was 5.5 months, median OS was not reached, and 12-month OS was 58%. Neither trial was associated with unexpected or unacceptable toxicities [71]. These two studies resulted in FDA approval of pembrolizumab in treatment of rrPMBCL that had relapsed following two lines of chemotherapy.

##### Renal Cell Carcinoma 

Following the success of a phase Ib investigation evaluating safety and efficacy of combination pembrolizumab/axitinib [72], the phase III KEYNOTE-426 compared combination pembrolizumab/axitinib to first-line sunitinib in patients with treatment-naïve advanced RCC. Results published earlier this year showed significantly longer OS in the pembrolizumab/axitinib group (HR = 0.53, *p* < 0.0001), as well as longer PFS (15.1 months vs. 11.1 months; HR = 0.69, *p* < 0.001). ORR was also significantly higher in the combination group (59.3% vs. 39.7%; *p* < 0.001) [73]. These results formed the basis of the recent FDA approval of pembrolizumab as first-line treatment in patients with advanced disease. 

### 3.4. PD-L1 Inhibitors 

#### 3.4.1. Atezolizumab

##### Overview 

Table A5 summarizes current clinical data for PD-L1 inhibitors.

##### Bladder Cancer

The IMvigor210 [74] and subsequent IMvigor211 [75] trials were the bases for initial FDA approval and expanded indications, respectively, of atezolizumab in treatment of urothelial cancer. IMvigor210 investigated atezolizumab monotherapy in patients with platinum-resistant, locally advanced or metastatic urothelial carcinoma; the follow up phase III IMvigor211 trial compared atezolizumab to various chemotherapies (investigator’s choice of vinflunine, paclitaxel, or docetaxel) in a similar patient population. Phase II data from IMvigor210 showed superior ORR compared to historical chemotherapy ORR (15% vs. 10%; stratified response rates to atezolizumab were as high as 26% in tumors with >5% PD-L1-positive cells) [74], though this significant improvement in ORR was not reproduced on direct comparison of chemotherapy to atezolizumab in IMvigor211. Tolerability data in the phase III study strongly favored atezolizumab, however, as grade 3–4 AE rate in the atezolizumab group was 20% versus 43% in chemotherapy groups [75]. The current FDA-approved indications are for treatment of patients with locally advanced or metastatic disease who are (A) ineligible for cisplatin-based therapy and have biochemical evidence of >5% PD-L1 expression within the tumor, or (B) ineligible for any platinum-based therapy.

##### Breast Cancer (Triple-Negative) 

In 2018 Schmid et al. published findings from a phase III trial in which patients with therapy-naïve triple-negative breast cancer (TNBC) were treated with either combination atezolizumab/nab-paclitaxel or nab-paclitaxel plus placebo. Median PFS was found to be significantly longer in the atezolizumab plus nab-paclitaxel group compared with the placebo plus nab-paclitaxel group (7.2 months vs. 5.5 months, HR = 0.80, *p* = 0.002); increased benefit was seen in subgroup analysis of those with baseline increased PD-L1 expression [76]. Thus, this combination was approved in March 2019 for patients with >1% PD-L1 expression.

##### Non-small Cell Lung Cancer 

In response to the POPLAR phase II trial, atezolizumab received FDA approval for treatment of metastatic NSCLC after progression on platinum-based chemotherapy. In this study, atezolizumab was compared to docetaxel in patients with previously treated advanced or metastatic NSCLC. Both OS (HR = 0.73) and AE profiles were superior in the atezolizumab group [77].

A second indication for atezolizumab was granted in December 2018 as part of first-line combination therapy with bevacizumab, paclitaxel, and carboplatin for metastatic NsqNSCLC. This was in response to the phase III IMpower150 trial, in which bevacizumab plus carboplatin and paclitaxel (BCP) was compared to atezolizumab plus bevacizumab, carboplatin, and paclitaxel (ABCP) for patients with previously untreated metastatic NsqNSCLC. Median PFS and OS were significantly longer in the ABCP group compared to the BCP group (PFS: 8.3 months vs. 6.8 months; HR = 0.62, *p* < 0.001; OS: 19.2 months vs. 14.7 months; HR = 0.78, *p* = 0.02), and ABCP had comparable tolerability to that of each independent agent [78].

##### Small Cell Lung Cancer 

In 2018, Horn et al. published results of the IMpower133 phase III trial comparing atezolizumab plus carboplatin and etoposide (ACE) to placebo plus carboplatin and etoposide (CE) in patients with extensive SCLC and no previous systemic therapy. Compared to CE, ACE had significantly longer OS (12.3 months versus 10.3 months; HR = 0.70, *p* = 0.0069) and PFS (5.2 months vs. 4.3 months; HR = 0.77, *p* = 0.0170), with a well-tolerated AE profile [79]. This led to an FDA approval in March 2019 as part of first-line combination treatment (ACE) in extensive-stage SCLC.

#### 3.4.2. Avelumab 

##### Bladder Cancer

Results from a phase Ib trial investigating the safety and efficacy of avelumab in patients with refractory metastatic urothelial carcinoma were published in 2017, demonstrating an ORR of 18.2%, with a CRR of 11.2% and 12-month OS rate of 54.3% [80]. This data supported avelumab’s excellent efficacy in advanced bladder cancer, leading to accelerated FDA approval in May 2017 for treatment of metastatic urothelial carcinoma refractory to 12 months of platinum-based chemotherapy. 

##### Merkel Cell Carcinoma 

Avelumab was granted its first FDA approval in March 2017 for the treatment of metastatic MCC in patients 12 years and older. This was in response to the findings from the JAVELIN Merkel 200 phase II trial evaluating avelumab monotherapy in patients with stage IV, chemotherapy-resistant, MCC. In this study, avelumab was associated with an ORR of 31.8% and a 9% CRR, as well as a favorable safety profile [81]. 

##### Renal Cell Carcinoma 

In a recent phase III trial, avelumab in combination with axitinib was compared to sunitinib monotherapy for first-line treatment of RCC. A clinically significant increase in PFS (HR = 0.61, *p* < 0.001) and ORR was demonstrated in the combination avelumab/axitinib group compared to the sunitinib group. In response to these results, this combination was approved for first-line treatment in patients with advanced RCC in May 2019 [82]. 

#### 3.4.3. Durvalumab

##### Bladder Cancer 

Durvalumab was granted accelerated FDA approval in May 2017 for the treatment of patients with locally advanced or metastatic urothelial carcinoma that had progressed during or following platinum-based chemotherapy and/or within 12 months of neoadjuvant or adjuvant treatment with platinum-based chemotherapy. This approval followed the results from a phase I/II study published by Massard et al. which demonstrated an ORR of 31% in patients with metastatic bladder cancer who had progressed on, been ineligible for, or refused any number of prior therapies [83]. 

##### Non-Small Cell Lung Cancer 

In 2017, Antonia et al. published results from the PACIFIC phase III trial comparing durvalumab to placebo as consolidation therapy in patients with stage III NSCLC that did not progress after two or more cycles of platinum-based chemoradiotherapy. Median PFS was significantly longer in the durvalumab group compared to placebo (16.8 months vs. 5.6 months; HR = 0.52, *p* < 0.001), and AE profile was acceptable [84]. This led to FDA approval in February 2018 for consolidation therapy in patients with unresectable stage III NSCLC without progression during concurrent platinum-based chemotherapy. 

### 3.5. LAG-3 Inhibitors 

#### 3.5.1. Relatlimab

In 2017, Ascierto et al. published preliminary phase I/IIa data from patients treated with combination relatlimab/nivolumab for melanoma that had progressed on prior anti-PD-1/anti-PD-L1 therapy. ORR for all patients was 11.5% (1 CR, 6 PR), with subgroup analyses demonstrating an ORR of 18% in patients with tumoral LAG-3 expression ≥1% [24]. Early data from a separate phase I study (NCT02658981) investigating relatlimab monotherapy and combination nivolumab/relatlimab for recurrent glioblastoma multiforme (GBM) showed excellent tolerability of relatlimab alone, with no dose-limiting toxicities (DLT) reported at the maximum planned dose [85].

#### 3.5.2. Ongoing Studies

Several clinical trials are underway investigating the role of anti-LAG-3 therapies in various malignancies. These are presented in Table A6.

### 3.6. TIM-3 Inhibitors 

As of September 2019, no clinical data has been published for TIM-3 inhibitors. Ongoing studies are summarized in Table A6.

### 3.7. CD40 Agonists 

#### 3.7.1. APX0050M 

##### Melanoma

APX005M in combination with nivolumab is under current exploration in the treatment of metastatic melanoma. New clinical data was recently published from a phase Ib dose-escalation/phase II dose-expansion trial investigating this combination in patients with metastatic melanoma that had progressed on anti-PD-1 monotherapy. A good safety profile was reported; phase II data showed partial response in two of 12 subjects, along with stable disease in three subjects [86].

##### Pancreatic Cancer 

Early results from an ongoing phase Ib study investigating APX005M in combination with chemotherapy (gemcitabine and nab-paclitaxel) and nivolumab for previously untreated metastatic pancreatic ductal adenocarcinoma were published in July 2019, with evidence of tolerable AE profiles and favorable anti-tumor activity of APX005M at varying doses and combinations [87]. A randomized phase II is in the works to expand upon the groundwork laid by this study.

#### 3.7.2. CP-870,893 

##### Metastatic Melanoma 

In 2015, Bajor et al. published the results of a phase I dose escalation study investigating combination CP-870,893/tremelimumab in patients with metastatic melanoma. A total of 24 patients were enrolled; four different dose combinations were employed. Three DLTs were reported. The maximum tolerated doses (MTD) were found to be 0.2 mg/kg of CP-870,893 and 10 mg/kg of tremelimumab. ORR was 27.3%, CRR was 9.1%, and PRR was 18.2%. Median PFS was 22 months and median OS was 26.1 months [88].

##### Pancreatic Cancer 

In November 2013, Beatty et al. published results of a phase I study, which evaluated the MTD, safety profile, and efficacy of combination CP-870,893/gemcitabine for the treatment of advanced pancreatic ductal carcinoma. This combination was tolerated well, with only one DLT. Four patients achieved partial response to treatment [89]. 

#### 3.7.3. Ongoing Studies

Other ongoing clinical studies evaluating CD40 agonists are summarized in Table A7.

### 3.8. OX40 Agonists 

#### 3.8.1. 9B12 

##### Advanced Solid Malignancies 

Safety and efficacy data for 9B12 was published for patients with various metastatic solid malignancies refractory to conventional chemotherapy [90,91,92]. The safety profile was acceptable, and regression of at least one tumor nodule was seen in 12 of 30 patients (40%).

#### 3.8.2. Ongoing Studies

Current investigations in OX40 agonist therapies are highlighted in Table A7.

### 3.9. Combination Therapies

As the research, development, and approval of checkpoint modulators has expanded, combination therapy, both with other checkpoint inhibitors as well as with traditional chemotherapy and radiotherapy, has garnered much therapeutic interest. Many of these combinations have been discussed in the sections above. As of September 2019, there are a total of 2250 active trials investigating the use of anti-PD-1/PD-L1 mAbs in various diseases, and of those, 1716 are testing combination therapies with other antineoplastic agents [93].

Combination therapies are studied in first-line settings with the intention of improving response rates to existing PD-1/PD-L1 monotherapy and to prevent activation of resistance pathways, ensuring durability of responses. The first combination therapy to be approved by the FDA was that of ipilimumab and nivolumab in 2015 for BRAF V600 wild type melanoma. This combination also gained approval for use in RCC and MSI-H CRC. The second combination to receive FDA approval was that of pembrolizumab and conventional chemotherapy in May 2018 for use in NsqNSCLC. Soon after, in October 2018, this combination gained approval in treating sqNSCLC and then again in June 2019 for treatment of patients with HNSCC. Additionally, pembrolizumab and avelumab have been approved in combination with the tyrosine kinase inhibitor axitinib for first-line treatment of advanced RCC. Most recently, the combination of pembrolizumab and the angiogenesis inhibitor lenvatinib (Lenvima, Eisai Co.) was granted approval in September 2019 for use in advanced endometrial carcinoma that has progressed on prior systemic therapy, is not MSI-H or dMMR, and is not amenable to curative surgery or radiation [94]. 

Combination therapies are also being evaluated in the PD-1/PD-L1-refractory setting as illustrated above, with a few combinations (PD-1 mAb + CD40 agonist, PD-1 mAb + LAG-3 inhibitor) showing promising initial data in terms of ORR and disease control rates [24,86,88]. Further long-term data from these studies would be extremely helpful, as PD-1/PD-L1 mAb resistance is a major therapeutic concern now. Some of these combinations (PD-1 antibody + LAG-3 inhibitor) are even being evaluated in the first-line setting based on the preliminary data noted in refractory setting (Trial NCT03743766).

## 4. Immunotherapy Resistance and Its Implications 

Despite the success of checkpoint immunomodulation, not all populations benefit. Resistance is well-documented, acting as a barrier to a durable (and/or early) tumor response [95,96,97,98].

Resistance is categorized into two classifications: primary and secondary. Primary resistance is defined by absent tumor response to initial therapy. This is in contrast to secondary resistance, in which originally susceptible tumor cells adapt over time to immunotherapy. Mechanisms of resistance are numerous; both tumor-cell-intrinsic and tumor-cell-extrinsic sources have been described. Similar methods are demonstrated in both primary and secondary resistance. Examples of tumor-cell-intrinsic mechanisms include lack of neoantigen development, impaired antigen processing and presentation, altered intracellular signaling pathways, and upregulated or constitutive expression of inhibitory ligands (Figure 6) [49,99,100,101,102,103,104]. Tumor-cell-extrinsic mechanisms include increased recruitment and activity of inhibitory immune cells within the TME and upregulation of LAG-3, TIM-3, and other inhibitory ligands (Figure 7) [27,30,97,105,106,107].

Several strategies are currently being researched to combat resistance to immunotherapy, including combination therapy with other checkpoint inhibitors, chemotherapy and/or radiation. This has been discussed above in Section 3.9. Generally, a single immune checkpoint inhibitor—most frequently an anti-PD-1 mAb—is employed with one or more additional therapies [97,98]. This framework is evident in many of the studies discussed previously, with a number of FDA approvals granted in response to the superior tumoral response and/or survival rates. Investigatory studies have shown promise for combination anti-PD-1 + anti-LAG-3 and anti-PD-1 + CD40 agonist therapies in patients with known PD-1/PD-L1-refractory disease states. In general, response rates tend to improve with combination therapy, but tolerability and AE profile worsens. Thus, a careful assessment of clinical benefit versus toxicity is necessary when evaluating the utility of combination therapies in different populations. 

The use of oncolytic viruses in conjunction with immunotherapy is an additional emerging strategy to fight resistance. Oncolytic viruses are genetically engineered viral strains designed to invade and lyse malignant cells without harming normal cells [108]. Furthermore, when injected locally, oncolytic viruses can change immune-secluded “cold” tumors to immune-rich “hot” tumors, which helps immunotherapy work more effectively. The first FDA-approved oncolytic virus, talimogene laherparepvec, was shown to be of benefit in the treatment of recurrent, unresectable stage IIIB-IVM1a melanoma [109]. Combination with immune checkpoint inhibitors has subsequently been explored. Thus far, early clinical data in advanced melanoma patients appear promising [110]. Investigations in treatment of other malignancies are ongoing. 

Furthermore, targeting specific components of the TME is another topic undergoing extensive investigation. Stromal cells within the TME have been implicated in multiple pro-neoplastic processes, including physical support, selective promotion of tumorigenesis, angiogenesis, tissue remodeling, and suppression of anti-tumor immunity [111,112,113]. Examples include myeloid-derived cell populations and cancer-associated fibroblasts (CAF).

Somewhat paradoxically, infiltration of the TME by various myeloid lineages has been associated with inhibited anti-tumor immunity, pro-tumor effects, and poorer overall prognoses [111]. Most commonly implicated cells include tumor-associated macrophages, myeloid-derived suppressor cells, tumor-associated dendritic cells, and tumor-associated neutrophils. The pro-malignancy effects are thought to be partially due to the plastic nature of the myeloid lineages; in response to various elements of the TME, including hypoxia, endoplasmic reticulum stress, and local immunosuppressive cytokines, these immune cells become polarized into immunotolerant phenotypes (M2, N2, *et cetera*) [114,115,116,117]. This, in turn, upregulates the expression of pro-angiogenic genes (*VEGF*), increases immunosuppressive and tissue remodeling cytokines (IL-10, transforming growth factor-beta (TGF-β)), suppresses pre-activated T cell proliferation, enhances recruitment and proliferation of Tregs and other immunotolerant myeloid lineages (macrophage colony-stimulating factor (M-CSF), CCL2), promotes Th2 differentiation while suppressing Th1 responses (PGE2), and decreases cancer neo-antigen expression by APCs [118,119,120,121,122,123]. Understandably, the prospect of limiting these effects is appealing in cancer therapy, and depletion of myeloid cells has been shown in mice to correlate to decreased tumor growth [113]. Broadly, two strategies have been investigated in pre-clinical models: depletion of myeloid cell numbers (via blockade of CCL2/CCR2, M-CSF/M-CSF receptor, and VEGF/VEGF receptor) and manipulation of myeloid cell function/plasticity (introduction of anti-cancer cytokines to TME, inhibition of STAT3 signaling, triggering toll-like receptors 3 and 9, and inhibiting CD36, inducible nitric oxide synthase, arginase-1, and indoleamine 2,3 deoxygenase) [113]. Results thus far have shown therapeutic benefit with both approaches; however, more research is necessary at this time to determine optimal targets. Addressing the problem of myeloid cells may prove to be helpful in reducing checkpoint modulator resistance as these potential therapies transition from the pre-clinical to clinical environment in the coming years.

Similarly, CAFs represent other troublesome inhabitants of the TME; they are also associated with significant immunosuppressive effects. A fairly heterogenous group with varying traits, CAFs typically have high metabolic activity, owing to their increased synthetic capacity compared to the traditionally indolent “normal” fibroblasts [112]. Many types of CAFs produce multiple growth factors and cytokines, including TGF-β, VEGF, and IL-6, which assist in immune evasion [124]. Some CAFs are also known for their extensive production of extracellular matrix, which blocks access of immune cells to the tumor. Therapies directed at CAFs thus far are primarily in pre-clinical stages, though some early clinical studies are ongoing. Mechanisms explored include inhibition of CAF function (TGF-β inhibitors, Hedgehog inhibitors, CXCR4 inhibitors), reprogramming CAFs to normal fibroblasts (vitamin A, D), and depletion of stromal extracellular matrix (anti-tenascin inhibitor). For certain CAF subtypes (fibroblast activation protein+), direct elimination via transgenic techniques and oncolytic viruses is also under investigation [112]. Early findings are promising, but additional research is necessary to further the understanding of these problematic TME inhabitants as a means to better select the most appropriate targets in this context.

## 5. Conclusions

Immune checkpoint modulators have garnered significant attention in the management of solid malignancies over the past several years mainly due to fewer side effects than chemotherapy, as well as their ability to result in durable responses in certain patients. The advent of these therapies has significantly changed treatment paradigms and prognoses of several cancers. Given these many advances, more and more immunomodulatory therapies are being explored, and those with pre-established indications are undergoing further investigations to expand their footprint within the oncologist’s armamentarium. Nonetheless, many patients do not respond to immunotherapy, and resistance proves challenging. As we explore strategies to combat resistance to immunotherapies, continued research to identify biomarkers to predict response and side effects of immunotherapy is crucial.

## Figures and Tables

**Figure 1 antibodies-08-00051-f001:**
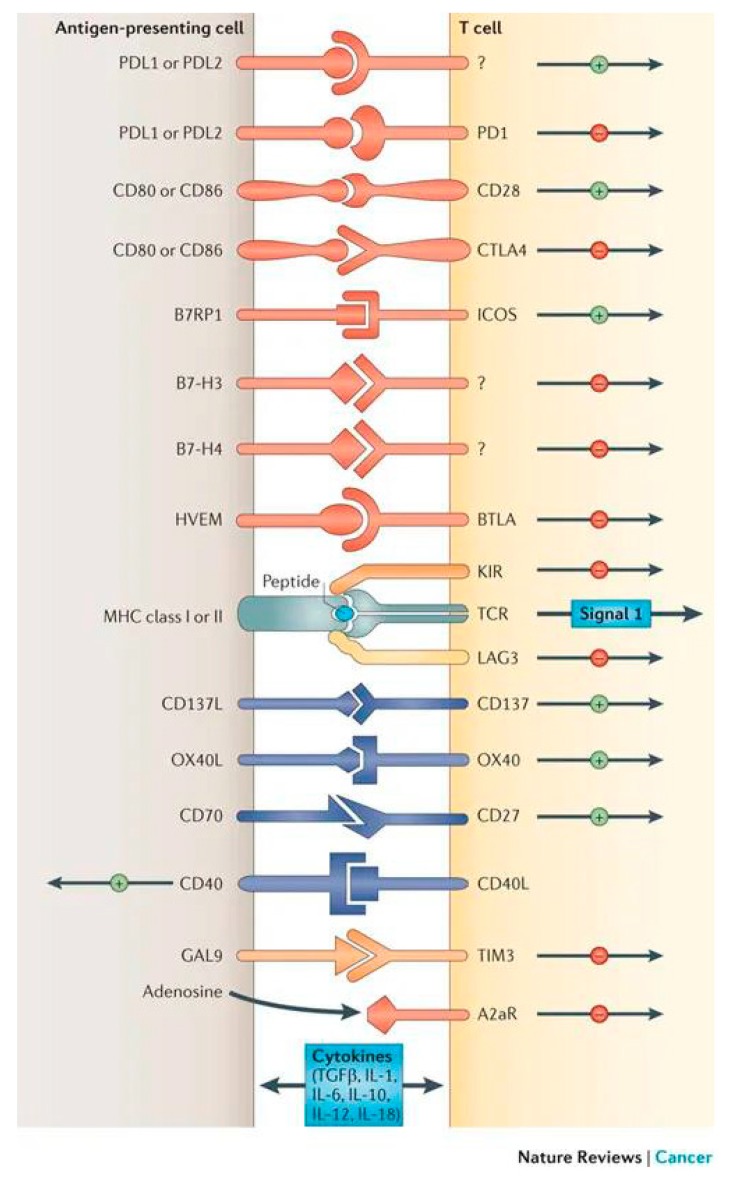
Overview of receptor-ligand interactions involved in checkpoint modulation. Description: Examples of different receptors and ligands involved in checkpoint modulation, along with generalized stimulatory (+) or inhibitory (−) effects. *Reprinted with permission from:*
**Spinger Nature:** Pardoll, D.M. The blockade of immune checkpoints in cancer immunotherapy. *Nat. Cancer Rev.*
**2012**, *12*, 252–264.

**Figure 2 antibodies-08-00051-f002:**
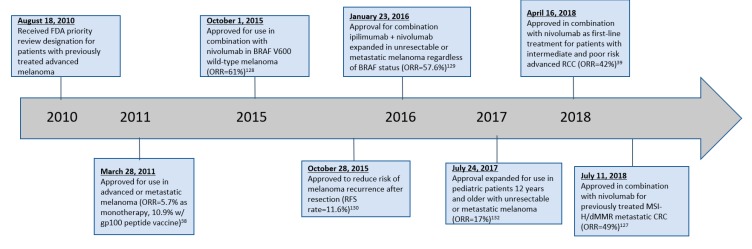
Timeline of ipilimumab FDA approvals.

**Figure 3 antibodies-08-00051-f003:**
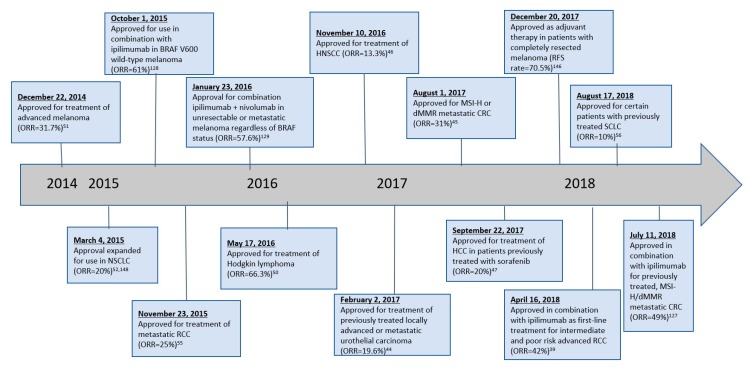
Timeline of nivolumab FDA approvals.

**Figure 4 antibodies-08-00051-f004:**
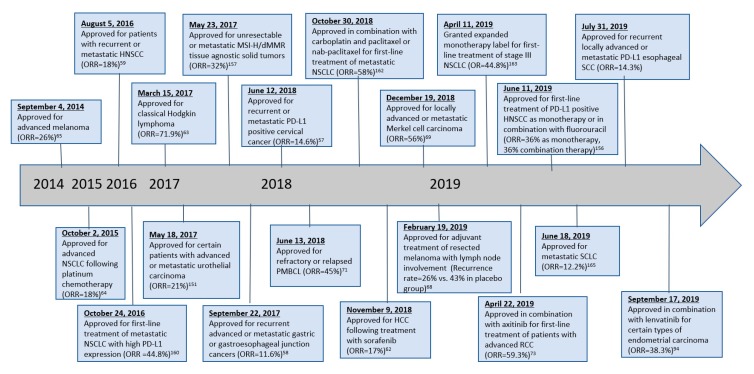
Timeline of pembrolizumab FDA approvals.

**Figure 5 antibodies-08-00051-f005:**
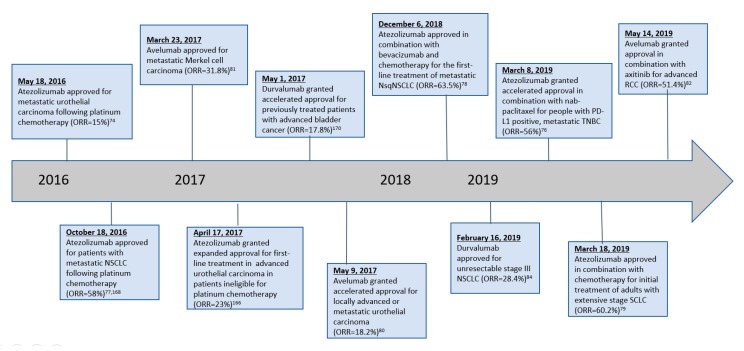
Timeline of PD-L1 inhibitor FDA approvals.

**Figure 6 antibodies-08-00051-f006:**
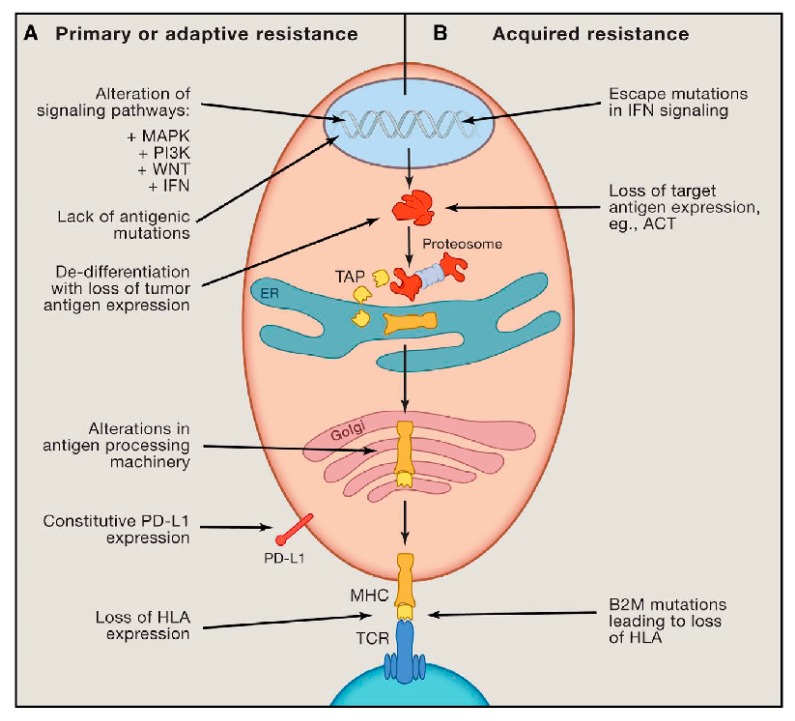
Intrinsic mechanisms of resistance to immunotherapy. *Description*: (**A**) Examples of intrinsic mechanisms of adaptive resistance, including altered signaling pathways, limited mutational burden, de-differentiation of tumor resulting in a loss of neoantigen expression, defective antigen processing, constitutive PD-L1 expression, and loss of HLA expression. (**B**) Examples of intrinsic mechanisms of acquired resistance, including loss of antigenic target, loss of HLA expression, and escape mutations in IFN signaling [97].

**Figure 7 antibodies-08-00051-f007:**
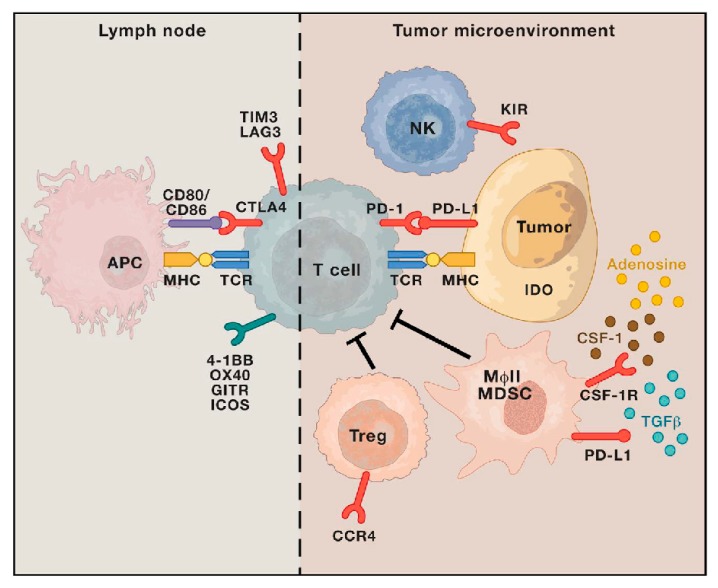
Extrinsic mechanisms of resistance to immunotherapy. *Description*: Examples of extrinsic mechanisms of resistance, including upregulated or constitutive immune checkpoint expression, immunosuppressive cytokine release (CSF-1, TGFβ, adenosine) within the tumor microenvironment, T cell exhaustion and phenotypic alteration, and increased immunosuppressive cell populations (Treg, MDSC, MɸII) [97].

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
