# Peer review of "Therapeutic Monoclonal Antibodies Targeting Immune Checkpoints for the Treatment of Solid Tumors"

_2073-4468, 2019, doi:10.3390/antib8040051_

Round 1

Reviewer 1 Report

I thank the authors for a good overview of a very long list of antibodies against checkpoint inhibitors that are currently in the clinic.

There are a few things that could be added/corrected.

In line 75 the authors state that the mechanism of action for CTLA-4 inhibitors is solely mediated by how they block CTLA-4 from interacting with B7-1 and B7-2, which then allows for CD28 to bind and activate T-cells. I believe that the jury is still out on the exact mechanism. A recent paper by Vargas et al. titled "Fc Effector Function Contributes to the Activity of Human Anti-CTLA-4 Antibodies" published in Cancer Cell in 2018 showed that FcgR binding of ipilimumab played a role in their function, hence the mechanism might be a bit more complicated. The authors should state clearly that the full mechanism for some of these targets is not well understood and still the subject of extensive investigation.

There are no figures in the paper. The authors could add a few figures such as a timeline for the testing of antibodies against immune checkpoints, highlighting the approval dates for the three targets, and the clear expansion of targets that are currently under investigation. This would help as I not always saw a consistent description of the chronology of the approvals for each drug.

A figure describing the different mechanisms of actions of the different targets would also be useful for the reader.

A figure of the different resistance mechanisms would also be helpful to follow section 4.

One thing that is still not clear to me after reading the review, is what are the differences between the different antibodies against the same target? In table 1 the authors could brake down further the differences of ipilimumab, tremelimumab, etc. What are the subtypes (IgG1 vs IgG2 vs IgG4) for each of them? What are the known affinities to their targets? Do all of them bind the same epitope? (Do they compete against each other?) Why would a doctor administer Nivolumab vs Pembrolizumab? 

Also, for Table 1, the authors should either remove the targets not still approved (LAG-3, TIM-3, CD40, OX40) or add a long list of other targets under investigation (41BB, iCOS, TIGIT, B7-H3/H4, etc etc). What is the rationale for including some and not others?

Between sections 3 and 4, the authors could add a section describing what are the most promising combinations that are ongoing at the moment. In a recent review (Nature Reviews Drug Discovery volume17pages854–855 (2018)) there is an extensive list of ongoing trials for combinations of anti-PD-1 and anti-PD-L1 antibodies with a large number of targets. This should be discussed in a separate section, or at least mentioned at the end of section 3.

I found it a bit confusing that the tables for the FDA-approved molecules came later than the tables for the not-yet approved molecules. The authors might want to change the order. It is clear that it was a lot of work to put together all the tables for the ongoing trials, which will be helpful to have in one review.

Author Response

The authors would like to thank the reviewer for taking their time to review this manuscript. We greatly appreciate the feedback and comments for improvements and have responded to them below in red.

Reviewer 1

I thank the authors for a good overview of a very long list of antibodies against checkpoint inhibitors that are currently in the clinic.

There are a few things that could be added/corrected.

In line 75 the authors state that the mechanism of action for CTLA-4 inhibitors is solely mediated by how they block CTLA-4 from interacting with B7-1 and B7-2, which then allows for CD28 to bind and activate T-cells. I believe that the jury is still out on the exact mechanism. A recent paper by Vargas et al. titled "Fc Effector Function Contributes to the Activity of Human Anti-CTLA-4 Antibodies" published in Cancer Cell in 2018 showed that FcgR binding of ipilimumab played a role in their function, hence the mechanism might be a bit more complicated. The authors should state clearly that the full mechanism for some of these targets is not well understood and still the subject of extensive investigation.

We have added a statement in Section 2.1 (Overview) that discusses that the full mechanisms of action are still being determined, and as such, what’s described touches on the more foundational aspects of our current understanding.

There are no figures in the paper. The authors could add a few figures such as a timeline for the testing of antibodies against immune checkpoints, highlighting the approval dates for the three targets, and the clear expansion of targets that are currently under investigation. This would help as I not always saw a consistent description of the chronology of the approvals for each drug.

We have added timelines of important FDA approvals for ipilimumab (Figure 2), nivolumab (Figure 3), pembrolizumab (Figure 4), and PD-L1 inhibitors (Figure 5) to provide a clearer description of the sequence of approvals over the years.

A figure describing the different mechanisms of actions of the different targets would also be useful for the reader.

We have requested permission for the use of a figure (Figure 1) from a prior Nature Reviews Cancer article that provides an illustration of numerous receptor-ligand interactions implicated in immune checkpoint modulation. This also shows simply whether each interaction results in immunostimulatory or immunosuppressive effects.

A figure of the different resistance mechanisms would also be helpful to follow section 4.

We have obtained permission for two figures (Figures 6 and 7) from a prior Cell review that illustrates mechanisms of intrinsic resistance (Figure 6) and extrinsic resistance (Figure 7).

One thing that is still not clear to me after reading the review, is what are the differences between the different antibodies against the same target? In table 1 the authors could brake down further the differences of ipilimumab, tremelimumab, etc. What are the subtypes (IgG1 vs IgG2 vs IgG4) for each of them? What are the known affinities to their targets? Do all of them bind the same epitope? (Do they compete against each other?) Why would a doctor administer Nivolumab vs Pembrolizumab? 

We have modified Table 1 to include IgG subtypes, affinity, and epitopes of each of the mAbs discussed. We have also added short paragraphs to the text that discuss these properties as well.

Also, for Table 1, the authors should either remove the targets not still approved (LAG-3, TIM-3, CD40, OX40) or add a long list of other targets under investigation (41BB, iCOS, TIGIT, B7-H3/H4, etc etc). What is the rationale for including some and not others?

We have removed LAG-3, TIM-3, CD40, and OX40 from Table 1.

Between sections 3 and 4, the authors could add a section describing what are the most promising combinations that are ongoing at the moment. In a recent review (Nature Reviews Drug Discovery volume17, pages854–855 (2018)) there is an extensive list of ongoing trials for combinations of anti-PD-1 and anti-PD-L1 antibodies with a large number of targets. This should be discussed in a separate section, or at least mentioned at the end of section 3.

We have provided a succinct discussion of combination therapies that have gained FDA approval previously, as well as briefly touched on the prevalence of combination therapy as it pertains to active clinical studies.

I found it a bit confusing that the tables for the FDA-approved molecules came later than the tables for the not-yet approved molecules. The authors might want to change the order. It is clear that it was a lot of work to put together all the tables for the ongoing trials, which will be helpful to have in one review.

We have reorganized the tables such that the LAG-3/TIM-3 table and CD40/OX40 table follow the respective tables for the mAbs with FDA-approved indications. Tables have been renumbered to reflect these changes.

Reviewer 2 Report

Conventional cancer therapy is a highly profitable business. Accordingly, it is of significant interest that the authors have indicated the "owners" of the various immune checkpoint modulators in the manuscript. For sake of a comprehensive overview, I suggest to include the respective institutions in table 1 as well.

Author Response

The authors would like to thank the reviewer for taking their time to review this manuscript. We greatly appreciate the feedback and comments for improvements and have responded to them below in red.

Reviewer 2

Conventional cancer therapy is a highly profitable business. Accordingly, it is of significant interest that the authors have indicated the "owners" of the various immune checkpoint modulators in the manuscript. For sake of a comprehensive overview, I suggest to include the respective institutions in table 1 as well.

We have added the respective institutions to the mAbs discussed in Table 1

Reviewer 3 Report

The Review by Gravbrot et al., "Therapeutic monoclonal antibodies targeting immune checkpoints for the treatment of solid tumors" provides a nice overview on current standard of care and early stage CPIs as well as CD40 and OX40 agonists.

The review is well structured as well as written and will only require minor stylistic and language editing.

The abstract does overpromises, as the section relating to: "...Furthermore, we review the challenges to durable tumor responses that are seen in some patients and discuss possible interventions to circumvent these barriers." is rather thin. I would strongly recommend to rework and expand on this section as well as add in additional examples of successful clinical CPI combinations, beyond oncolytic viruses. Furthermore the authors should discuss the crucial importance of stromal elements, such as e.g. cancer associated fibroblasts (CAFs) and in particular the myeloid tumor compartment (e.g. TAMs) in more detail.

For Ipilimumab and anti-CTLA-4 therapeutics in general the authors should discuss Tregs in more detail and also relate to potential Treg depletion (ADCC) in e.g. pre-clinical tumor models and Melanoma patients.

For anti-PD-1 vs. anti-PD-L1 it is important to highlight the key MoA differences, such as the PD-L2 interaction.

Besides an expansion of section 4 (see above) the manuscript would greatly benefit form a graphical summary of ORRs for the various CPIs in the different indications (at least anti-CTLA-4/-PD-1 and -PD-L1). Furthermore a graphical depiction of the ORRs in indication, for which CPIs are approved vs. all indications in which CPIs have been tested would be very helpful.

Other than that a very helpful review

Author Response

The authors would like to thank the reviewer for taking their time to review this manuscript. We greatly appreciate the feedback and comments for improvements and have responded to them below in red.

Reviewer 3

The Review by Gravbrot et al., "Therapeutic monoclonal antibodies targeting immune checkpoints for the treatment of solid tumors" provides a nice overview on current standard of care and early stage CPIs as well as CD40 and OX40 agonists.

The review is well structured as well as written and will only require minor stylistic and language editing.

The abstract does overpromises, as the section relating to: "...Furthermore, we review the challenges to durable tumor responses that are seen in some patients and discuss possible interventions to circumvent these barriers." is rather thin. I would strongly recommend to rework and expand on this section as well as add in additional examples of successful clinical CPI combinations, beyond oncolytic viruses. Furthermore the authors should discuss the crucial importance of stromal elements, such as e.g. cancer associated fibroblasts (CAFs) and in particular the myeloid tumor compartment (e.g. TAMs) in more detail.

We have expanded this section to include a discussion on past combination therapies with reported clinical success. We have also given examples of current combination therapies employed in PD-1/PD-L1-resistant populations that show promise thus far based on initial data. This discussion can be found in Section 3.9. We also have added a section on tumor microenvironment elements in Section 4.

For Ipilimumab and anti-CTLA-4 therapeutics in general the authors should discuss Tregs in more detail and also relate to potential Treg depletion (ADCC) in e.g. pre-clinical tumor models and Melanoma patients.

We have added further discussion about the role of Tregs and Treg depletion in Section 2.2.

For anti-PD-1 vs. anti-PD-L1 it is important to highlight the key MoA differences, such as the PD-L2 interaction.

We have briefly highlighted differences between PD-1 and PD-L1 MOAs in Section 2.3 and 2.4.

Besides an expansion of section 4 (see above) the manuscript would greatly benefit form a graphical summary of ORRs for the various CPIs in the different indications (at least anti-CTLA-4/-PD-1 and -PD-L1). Furthermore a graphical depiction of the ORRs in indication, for which CPIs are approved vs. all indications in which CPIs have been tested would be very helpful.

ORRs have been added to Figures 2-5 (timelines of FDA approvals for respective mAbs/mAb classes). A separate column has been added to Table 1 specifically for indications with clinical data but no FDA approval. This is adjacent to the column with FDA-approved indications for each drug, which we believe serves as a convenient way to compare which disease states each drug has approved indications for vs. purely investigational status without FDA approval.

Other than that a very helpful review
